# The Nrf2 Pathway in Ischemic Stroke: A Review

**DOI:** 10.3390/molecules26165001

**Published:** 2021-08-18

**Authors:** Marcelo Farina, Leonardo Eugênio Vieira, Brigitta Buttari, Elisabetta Profumo, Luciano Saso

**Affiliations:** 1Department of Biochemistry, Federal University of Santa Catarina, 88040-900 Florianópolis, Brazil; leonardovieira175@gmail.com; 2Department of Cardiovascular, Endocrine-Metabolic Diseases, and Aging, Italian National Institute of Health, 00161 Rome, Italy; brigitta.buttari@iss.it (B.B.); elisabetta.profumo@iss.it (E.P.); 3Department of Physiology and Pharmacology “Vittorio Erspamer”, Sapienza University of Rome, 00185 Rome, Italy

**Keywords:** ischemic stroke, Nrf2, treatment, preclinical studies, oxidative stress, ischemic cascade

## Abstract

Ischemic stroke, characterized by the sudden loss of blood flow in specific area(s) of the brain, is the leading cause of permanent disability and is among the leading causes of death worldwide. The only approved pharmacological treatment for acute ischemic stroke (intravenous thrombolysis with recombinant tissue plasminogen activator) has significant clinical limitations and does not consider the complex set of events taking place after the onset of ischemic stroke (ischemic cascade), which is characterized by significant pro-oxidative events. The transcription factor Nuclear factor erythroid 2-related factor 2 (Nrf2), which regulates the expression of a great number of antioxidant and/or defense proteins, has been pointed as a potential pharmacological target involved in the mitigation of deleterious oxidative events taking place at the ischemic cascade. This review summarizes studies concerning the protective role of Nrf2 in experimental models of ischemic stroke, emphasizing molecular events resulting from ischemic stroke that are, in parallel, modulated by Nrf2. Considering the acute nature of ischemic stroke, we discuss the challenges in using a putative pharmacological strategy (Nrf2 activator) that relies upon transcription, translation and metabolically active cells in treating ischemic stroke patients.

## 1. Introduction

Ischemic stroke, the most common one, accounts for approximately 87% of all stroke cases. It is characterized by the sudden loss of blood flow caused by thrombosis or embolism that occludes cerebral vessel(s) supplying specific area(s) of the brain [1]. Ischemic stroke is the leading cause of permanent disability and is among the leading causes of death worldwide. Globally, one in six people will have a stroke in their lifetime and around 14 million have a stroke each year. Even though stroke was initially classified as a condition affecting blood vessels, it was recently reclassified and is currently considered a neurological disorder; this led to improvements in acute healthcare and acquisition of research funding for stroke [2].

There is a great number of modifiable and non-modifiable risk factors for ischemic stroke. Among the non-modifiable risk factors, the most important ones are (i) age (the incidence of stroke increases with age [3]), (ii) sex (incidence is greater at younger ages in women, but increases with older age in men [4]), (iii) ethnicity (Hispanic and black populations are at higher risk of stroke than white populations [5]) and (iv) genetics (parental stroke by 65 years of age was associated with a 3-fold increase in risk of offspring stroke [6]. Among the modifiable risk factors, the most important ones are (i) hypertension (high blood pressure is one of the predominant risk factors and a 10 mm Hg increase in systolic blood pressure has been associated with a 38% increased stroke risk [7]), (ii) hyperglycemia (impaired glucose tolerance is an independent risk factor for future stroke [8]), (iii) atrial fibrillation (contributes to 15% of all strokes [9]), (iv) hyperlipidemia (total plasma cholesterol is positively associated with risk of stroke, while plasmatic levels of high-density lipoprotein are negatively associated with risk of stroke [10]), (v) smoking (tobacco smoking is directly linked to increased risk of stroke [11]) and (vi) insufficient physical inactivity and poor diet (lack of exercise increases the chances of a stroke episode and poor diet influences the risk of stroke, contributing to hypertension, hyperlipidemia and diabetes [12,13]. In addition to the aforementioned factors, socioeconomic variation also affects the occurrence and/or outcomes of stroke; for broader information regarding risk factors for ischemic stroke, see reference [14].

Regarding treatments, intravenous thrombolysis with recombinant tissue plasminogen activator is the only approved pharmacological treatment for acute ischemic stroke. It has significant beneficial effects in acute ischemic stroke when administered between 3 and 4.5 h after the onset of symptoms [15], although an extension of this time interval is currently under debate [16]. This thrombolytic treatment aims at stimulating fibrinolysis to allow for clot removal, but does not consider the complex set of events taking place after the onset of ischemic stroke, called ischemic cascade (discussed below).

A great number of experimental studies has been performed in order to discover drugs able to mitigate the neurodegeneration following ischemic episodes. Of particular importance, oxidative stress has been highlighted as a potential pharmacological target in ischemic stroke, which is in line with the increased generation and decreased detoxification of oxidant molecules leading to stroke-mediated neurodegeneration [17]. Among the potential oxidative stress-related molecular targets, the transcription factor Nrf2 (Nuclear factor erythroid 2-related factor 2), which regulates the expression of a great number of antioxidant and/or defense proteins (discussed below), has been pointed as a potential pharmacological target involved in the mitigation of deleterious oxidative events taking place at the ischemic cascade. In this review, we summarized studies concerning the protective role of Nrf2 in experimental models of ischemic stroke, emphasizing molecular events resulting from ischemic stroke that are, in parallel, modulated by Nrf2. We also reviewed the available experimental literature concerning the effects of Nrf2 activators in ischemic stroke models, discussing the potential pharmacological use of Nrf2 activators in ischemic stroke patients.

## 2. The Ischemic Cascade and Oxidative Consequences

Ischemic stroke is characterized by the interruption or sudden restriction of cerebral blood flow in specific area(s) of the brain. Considering that brain metabolism is greatly dependent upon blood-derived glucose and oxygen, which allow for the proper functioning of glycolysis, tricarboxylic acid cycle and mitochondrial electron transport chain [18], ischemic stroke leads to major changes in cellular bioenergetics. The lack of proper adenosine triphosphate (ATP) levels represents a primary metabolic change resulting from ischemia; it is a main trigger for the initiation of a series of molecular deleterious events known as ischemic cascade, which is particularly detrimental to neurons due to its highly oxidative and glucose-dependent metabolism.

Because of the crucial role of ATP in maintaining cellular (especially, neuronal) ionic homeostasis, a significant ionic imbalance occurs few minutes after ischemia, with the abnormal influx of Na+ and efflux of K+, contributing to extensive depolarization and water transport into cells [19], which leads to cytotoxic edema. The low ATP synthesis, followed by Na+/K+ imbalance (due to Na+/K+ ATPase), also decreases the uptake of glutamate, the main excitatory neurotransmitter. This phenomenon is related to the fact that the action potential induced by glutamate on postsynaptic receptors is terminated by its clearance from the synaptic cleft by transporters located in neurons and, remarkably, in glial cells. Several glutamate transporters are dependent on extracellular Na+, thus on the activity of Na+/K+ ATPase. The impaired removal of glutamate, or even its release through the reverse operation of their transporters, represents an important event in the ischemic cascade, leading to neuronal toxicity due to excessive excitatory neurotransmission (excitotoxicity) [20]. In fact, increased glutamate levels in the synaptic cleft cause overstimulation of neuronal post-synaptic glutamate receptors, stimulating sodium and calcium influx. This may induce cytoplasmic calcium overload and activation of diverse enzymes, such as phospholipases, proteases and nucleases, which drive the breakdown of phospholipids, proteins and nucleases [20]. Of note, the depolarization of adjacent neurons produces a further calcium influx and additional glutamate release, leading to local amplification of the ischemic damage [21].

Cytoplasmic calcium overload is also detrimental to the mitochondrial function, being linked to the mitochondrial production of oxidant molecules [22]. Some mechanisms related to the pro-oxidative role of calcium in the ischemic cascade include a calcium-stimulated increase in metabolic rate, nitric oxide production and cardiolipin peroxidation. Thus, the excitatory overstimulation resulting from low ATP levels may culminate in oxidative damage, which represents a critical event leading to neuronal damage in this hypoxic phase of stroke [23]. In addition, it is important to mention that the synthesis of glutathione (GSH), a main low-molecular weight intracellular antioxidant, is dependent upon ATP. Consequently, decreased GSH synthesis subsequent to low ATP levels may also contribute to the redox imbalance and oxidative damage resulting from ischemia.

Even though fast reoxygenation, via reperfusion, is a desired step required to mitigate the metabolic stress that takes place in ischemic stroke, reoxygenation may also contribute to the generation of reactive oxidants [24], thus exacerbating the ischemia-reperfusion oxidative injury. Approximately 4 decades ago, considering the low levels of molecular oxygen in ischemic tissues, there was no reason to suppose that ischemia involved elevated production of oxygen-derived reactive species. However, evidence showed that a significant part of the damage resulting from ischemia may be more accurately called reperfusion injury or post-ischemic injury. Indeed, much of the injury was shown to occur not during the period of hypoxia but rather during the period when molecular oxygen is reintroduced to the tissue [25]. In this scenario, experimental evidence showing the protective effects of superoxide dismutase (SOD) indicated that superoxide was a critical molecule in ischemic (or post-ischemic) events [26,27]. Concerning the mechanisms mediating the generation of oxygen radicals in the ischemic cascade, a main source of superoxide in post-ischemic tissues is the enzyme xanthine oxidase, which is usually synthesized as a dehydrogenase (type D) and able to catalyze the conversion of xanthine into uric acid with no production of superoxide. However, under certain conditions (pro-oxidative environment and increased intracellular calcium levels), xanthine dehydrogenase is converted into xanthine oxidase in vivo in ischemic tissues, catalyzing the conversion of xanthine into uric acid with production of superoxide [28]. In addition, the ischemia-related depletion of ATP is paralleled by an increase in the levels of AMP, adenosine, inosine and hypoxanthine and xanthine. These two last products of purine catabolism represent the substrate for xanthine oxidase. The “new enzyme” with oxidase activity and the availability of molecular oxygen during reperfusion represent two important factors that contribute to the oxidative stress taking place during the post-ischemic (reperfusion/reoxygenation) period [25]. Currently, since most of the studies concerning ischemia-reoxygenation do not dissociate ischemic- from post-ischemic-related injuries, the term ischemic-reperfusion injury (IRI) has been used to correctly refer to the damage resulting from ischemia following reoxygenation and/or reperfusion.

During the last decades, evidence has highlighted additional molecules that stimulate oxidative damage toward biomolecules in IRI. One of these molecules is phospholipase A2 (PLA2), whose activation represents a critical metabolic event in ischemic stroke, leading to the hydrolysis of membrane phospholipids and consequent release of lysophopholipids and free fatty acids (FFAs), including arachidonic acid, a metabolic precursor for pro-inflammatory eicosanoids [29]. PLA2 (mitochondrial secretory isoform) also catalyzes the hydrolysis of cardiolipin, leading to disruption of the mitochondrial respiratory chain and increased production of reactive oxygen species (ROS) [30]. In addition, the oxidative metabolism of arachidonic acid also generates ROS. Both events contribute to the occurrence of lipid peroxidation, whose end products (i.e., malondialdehyde and 4-hydroxynonenal) covalently bind to proteins/nucleic acids, altering their function and causing cellular damage [29]. FFAs released in PLA2-catalyzed reactions can accumulate following ischemic stroke, undergoing oxidative metabolism by non-enzymatic and enzymatic processes catalyzed mainly by cyclooxygenases (COXs) and lipoxygenases (LOXs), resulting in the formation of lipid oxoderivatives [31], which modulate inflammatory and pro-oxidative processes. In line with this, experimental evidence indicates that both COXs [32,33] and LOXs [34,35] represent potential pharmacological targets for stroke therapy. This experimental (nonclinical) evidence has provided new insights into the regulation of inflammatory and pro-oxidative events in the ischemic brain. However, the potential translation of such experimental data to clinic scenarios remains a matter of debate [36].

In addition to COXs and LOXs, nicotinamide adenine dinucleotide phosphate (NADPH) oxidases (NOXs), a family of enzymes that catalyze the production of superoxide by transferring one electron from NADPH to molecular oxygen, have also been reported to exert detrimental effects on ischemic brain tissue. Evidence shows that NOX-knockout mice are resistant to damage due to experimental stroke and the infarct size and blood–brain barrier breakdown are enhanced in mice with pericyte-specific overexpression of NOX4 [37]. A recent experimental study has elegantly identified type 5 NADPH oxidase (NOX5) as a major player of IRI. Using in vitro organotypic cultures, the authors found that reoxygenation or calcium overload increased brain ROS levels in a NOX5-dependent manner. Based on in vivo approaches, the authors also showed that postischemic ROS formation, infarct volume and functional outcomes were worsened in NOX5-KI mice [38].

Based on the aforementioned evidence, it becomes clear that several pro-oxidative events display central roles for the occurrence of IRI. Some of these events include (i) calcium-mediated oxidative events (through increased metabolic rate, nitric oxide production and cardiolipin peroxidation), (ii) impaired GSH synthesis, (iii) increased superoxide generation (in dysfunctional mitochondria, as well as in xanthine oxidase and NOX-catalyzed reactions) and (iv) formation of lipid oxoderivatives (including in LOX- and COX-catalyzed reactions). The knowledge concerning the involvement of oxidative events in the ischemic cascade and consequently in IRI, was mostly derived from experimental studies (in vitro approaches and in vivo animal studies). Of note, such experimental studies have also elucidated several ischemia/reoxygenation-mediated biochemical and histological oxidative changes toward biomolecules, such as lipid peroxidation [39] and nucleic acid oxidation [40], as well as protein carbonylation [41] and nitrosylation [42]. Of note, increases in the levels of some of these oxidative-stress-related biomarkers have also been observed in patients with ischemic stroke [43], highlighting the significance of oxidative events in human IRI. Such findings suggest that drugs able to mitigate such oxidative events represent potential pharmacological strategies to treat ischemic stroke patients. Among the variety of molecules and/or targets that might be useful to mitigate IRI-mediated oxidative stress, there is the transcription factor Nrf2, which is the main topic of this review (Section 3).

## 3. Nrf2 Signaling Pathway and Ischemic Stroke

### 3.1. Overview of the Nrf2 Signaling Pathway

Nrf2 is a member of the cap’n’collar family of transcription factors and is present in various cell types. It consists of 605 amino acids with 7 highly conserved Nrf2-ECH domains (Neh1-7), which serve as a different functional region [44,45,46,47,48]. The Neh1 regulates DNA binding through the CNC–bZIP [49] and a nuclear localization signal (NLS) is responsible for the nuclear translocation of Nrf2 [50]. The Neh2, an N terminal regulatory domain, consists of DLG (low affinity) and ETGE (high affinity) motifs for the interaction with the Nrf2 negative regulator Kelch-like ECH-associated protein 1 (Keap1), which influences the stability and ubiquitination of Nrf2 [51]. The Neh3, Neh4 and Neh5 are transactivation domains mediating the interaction of Nrf2 with other coactivators [52,53], while the Neh5 domain is responsible for its cytoplasmic localization [54]. The Neh6 domain is a negative regulatory domain which binds to a β-transducin repeat-containing protein (β-TrCP), leading to Nrf2 ubiquitination, or regulates the Nrf2 stability by phosphorylation of serine residues [55]. The Neh7 domain inhibits the Nrf2-antioxidant response element (ARE) signaling pathway by promoting the binding of Nrf2 to the retinoic X receptor α (RXRα) [56].

In homeostatic conditions, Nrf2 stays in its inactive form within cells via Keap1. Keap1 is a cysteine-rich (27 cysteines), cytoplasmic, actin cytoskeleton-associated adapter zinc-metalloprotein of the Cul3/Rbx1 complex. It consists of five domains and Keap1/Cul3 homodimerization is regulated by the N-terminal portion of the intervening region with the BTB (Broad complex, Tramtrack and Bric-a-Brac) domain. The BTB domain of Keap1 plays a key role in sensing environmental electrophiles and is believed to be the target for several small molecule covalent activators of the Nrf2 pathway [57,58,59]. In the cytoplasm, Keap1 homodimerizes and binds to the cullin-based (Cul3) E3 ligase, forming Keap1-Cul3-RBX1 (Ring box protein-1) E3 ligase complex, that interacts with the Neh2 domain, forms the Keap1-Nrf2 complex and initiates degradation of Nrf2 by ubiquitination and proteasomal degradation [60,61,62].

In stress conditions (excessive accumulation of ROS, electrophilic molecules and proteotoxic stress), Nrf2 is released from the Keap1-Cul3-RBX1 complex and translocates into the nucleus, wherein it heterodimerizes with small Maf proteins (sMaf) and binds to the AREs on DNA, leading to the transcription of Nrf2 target genes [63]. The Nrf2/Keap1 pathway regulates a coordinated activation of a battery of cytoprotective genes that include biotransformation enzymes, antioxidant proteins, drug transporters, anti-apoptotic proteins and proteasome proteins. There are over 250 currently identified NRF2 target genes involved with redox regulation [44,58,64,65,66,67,68]. For example, target genes of Nrf2 are glutamate-cysteine ligase, NAD(P)H qui-none oxidoreductase 1 (NQO1), heme-oxygenase (HO-1), sulfiredoxin1 (SRXN1), heme-oxygenase (HO-1), glutathione S-transferase (GST), multidrug resistance-associated proteins (MRPs) and UDP-glucuronosyltransferase (UGT) [57]. Figure 1 depicts major molecules and events modulating Nrf2 stability and activation, as well as main downstream protein targets and their functions.

In addition, Nrf2 signaling takes part in the regulation of the cellular response to inflammation cooperating with NF-κB signaling pathways via suppression of pro-inflammatory genes, redox homeostasis and controls fundamental cellular processes, such as apoptosis, autophagy, angiogenesis, proliferation and cell migration [69,70]. Of note, Nrf2 can indirectly control the transcription of a host of non-ARE-containing genes. Indeed, functional AREs have been identified in the promoters of a number of transcription factors involved in DNA damage repair and apoptosis prevention [71,72,73].

Regulation of Nrf2 mainly occurs through the controlled maintenance of Nrf2 protein levels at the post-transcriptional and post-translational levels, as well as via epigenetic factors and interaction with other signaling pathways. It is important to note that its regulation mainly depends on the physiological and pathological context.

### 3.2. Molecular Events Linking Ischemic Stroke and Nrf2 Pathway

In a search using the terms “Nrf2” AND “ischemic stroke” in the PubMed database (https://pubmed.ncbi.nlm.nih.gov, accessed on 25 June 2021), it was possible to detect molecular players that are closely related to both topics. Among these players, there are several redox-active molecules (discussed below). Before discussing their potential (patho)physiological roles in the interplay between ischemic stroke and Nrf2 pathway, it is important to mention that most of the knowledge concerning molecular mechanisms involved in ischemic stroke and Nrf2 pathway comes primarily from experimental studies, including a great number of in vivo studies with rodents (mouse and rats). Of note, different experimental protocols have been developed to simulate different conditions, such as focal or global, as well as transient or permanent cerebral ischemia. Even though each of these protocols has specific features, the increased levels of reactive oxygen species and markers of oxidative damage are commonly observed in either focal or global, in addition to transient or permanent models [75,76,77,78]. Of note, data on the relationship between the Nrf2 pathway and ischemic stroke have been derived from these different experimental approaches (for a detailed review concerning such models, see reference [79]). Nonetheless, experimental ischemic models based on reperfusion/reoxygenation are expected to provide increased rates of oxidative stress considering the critical role of reoxygenation for the production of ROS [25].

As previously discussed (Section 2), increased levels of ROS are observed after either ischemia or ischemia/reoxygenation events. Hydrogen peroxide, a ROS whose levels are increased after ischemia/reoxygenation [27], is able to up regulate Nrf2 [80,81]. In agreement with this observation, experimental studies have reported the endogenous activation of Nrf2 following ischemic stroke, suggesting that this event represents a physiological response to the stress to which cells are subjected in ischemia/reperfusion (IR). Based on an IR protocol with a luciferase mouse model (a Keap1-dependent oxidative stress detector to visualize the Nrf2 expression from brain ischemia), Takagi and collaborators [82] showed increased levels of Nrf2 in the cerebral cortex and striatum of mice subjected to transient middle cerebral artery occlusion. The increased levels of Nrf2 were observed in both neurons and astrocytes and, notably, mainly in the penumbra zone. In another study based on an experimental model of transient cerebral ischemia [83], the authors reported the temporal and spatial distribution of Nrf2 in the nuclear and cytoplasmic compartments in cells in the ischemic core and peri-infarct regions and contralateral hemisphere of rat. Based on a quantitative immunohistochemical technique, these authors observed increased Nrf2 expression in brain sections in core and peri-infarct regions after 24 h reperfusion, with levels remaining elevated only in peri-infarct regions after 72 h. These two studies [82,83] provide evidence of Nrf2 activation following IR, suggesting that it represents an endogenous response resultant, at least partially, from ROS produced during transient ischemia.

On the other hand, several Nrf2-downstream proteins can mediate redox balance by neutralizing IR-derived ROS, such as superoxide anion and hydrogen peroxide. Particularly, superoxide dismutase and catalase, which metabolize superoxide anion and hydrogen peroxide, respectively, represent well-known Nrf2-downstream proteins [84,85,86,87]. Additional proteins, such as glutamate-cysteine ligase, glutathione peroxidase, glutathione reductase, thioredoxin reductase, heme oxygenase-1 and NADPH:quinone oxidoreductase, among others, are also known Nrf2-downstream molecules mediating redox balance and mitigating oxidative stress [74]. Table 1 presents experimental studies reporting the endogenous modulation of Nrf2 and/or its main Nrf2-downstream proteins after cerebral ischemia or ischemia-reperfusion. Although several studies presented in Table 1 were aimed at investigating protective effects of exogenously administrated Nrf2 activators against cerebral ischemic stroke, we have initially focused only on the potential endogenous modulation of Nrf2, evaluating the differences between control/sham and ischemic animals. It is noteworthy that most of these studies, which were based on either transient or permanent models of cerebral ischemia, reported significant increases in mRNA and/or protein expression of Nrf2 and Nrf2-downstream targets, indicating that the endogenous upregulation of Nrf2 represents an event resulting from cerebral ischemia or ischemia-reperfusion. Of note, some of these studies [88,89] indicated that Nrf2 knockout animals (Nrf2^−/−^) are more susceptible to cerebral ischemic stroke, highlighting the significance of this transcription factor in protecting the ischemic cerebral tissue. On the other hand, there are some studies (less frequent) reporting decreased gene and/or protein expression of Nrf2 and/or downstream proteins in the brain of ischemic animals, compared to controls. Although these apparent contradictory results may result from several causes, it is likely that the decreased gene and/or protein expression of Nrf2 and downstream proteins observed in ischemic animals in some studies can be related, at least partially, to an extreme rate of tissue damage, resulting in improper capability of performing transcription and translation.

As already discussed, ROS whose levels are increased after ischemia/reoxygenation, such as hydrogen peroxide [27], are able to upregulate Nrf2 [80,81]. On the other hand, there are specific Nrf2 downstream proteins capable of counteracting oxidants; this highlights an interesting interplay between ischemic stroke and Nrf2. From a Cartesian point of view, IR leads to increased levels of oxidants that, in turn, can activate Nrf2. On the other hand, Nrf2-downstream proteins can mitigate the deleterious effects of oxidants produced in excess during IR (see References from Table 1). Figure 2 depicts a schematic view of this interplay between ischemic stroke and Nrf2 pathway.

### 3.3. Effects of Nrf2 Modulators in Ischemic Stroke: Evidence from Experimental Studies

Given the pivotal role of Nrf2 in redox balance, several studies have reported its involvement in modulating cellular homeostasis in physiological and/or pathological conditions [64,147]. Particularly, promising neuroprotective effects of Nrf2 have been reported in diverse experimental models [148,149,150]. In the context of ischemic stroke, in addition to the protective effects exhibited by the endogenous activation of Nrf2 following ischemia or IR (discussed in Section 3.1, see Table 1), protective effects resulting from exogenously induced Nrf2 activation have also been reported, predominantly in rodent models (see examples discussed below). Of note, some Nrf2-activating compounds have displayed superior neuroprotective effects against IRI in wild-type compared to Nrf2^−/−^ animals [89,151], confirming the involvement of Nrf2 in mediating the beneficial effects of these molecules. Although there is a large number of molecules capable of activating Nrf2 and exhibiting protective effects in ischemic stroke models (described in the last paragraph of this section), we provide a more detailed discussion on the most frequently reported agents, as follows.

#### 3.3.1. Curcumin

Curcumin {1,7-bis(4-hydroxy-3-methoxyphenyl)-1,6-heptadiene-3,5-dione} (diferuloyl methane), a phytochemical compound extracted from *Curcuma longa* rhizomes, has been extensively studied for its multiple biological activities, including anti-inflammatory, antioxidant and anti-infective properties [152]. This polyphenol having a long history of use in traditional medicines of China and India, has a favorable safe profile. It is able to cross the BBB [153,154] with no toxicity, even at a high dose [155]. Curcumin has a Michael acceptor in the form of a α,β-unsaturated carbonyl group; thus, the main mechanism, by which it activates Nrf2, is by alkylating a protein thiol on the Keap-1-Nrf2 binding complex, which allows Nrf2 to translocate to the nucleus to initiate antioxidant gene expression changes [156,157]. Recently, mass spectrometric analysis revealed that curcumin binds to Keap1 Cys151 in the BTB domain, supporting that this amino acid is a critical target for curcumin modification of Keap1, which facilitates the liberation of Nrf2 [158]. Curcumin has been proven to exert neuroprotective effects and to prevent ischemic stroke through the attenuation of neurological dysfunction, infarction size, brain edema and BBB disruption [159,160,161,162,163,164,165,166,167,168,169] via anti-oxidant, anti-inflammatory and anti-apoptotic effects [170,171]. In vivo and in vitro [172,173], evidence demonstrates curcumin is an effective activator of Nrf2 in cerebral IR injury. The Nrf2/ARE signal pathway plays an important role at a very early time in rat brains subjected to middle cerebral artery occlusion (MCAO), a classic animal model of stroke. Indeed, both Nrf2 and HO-1 raise significantly in the first 3 h and maximize at 24 h after MCAO [174]. After systemic administration of curcumin, Nrf2 and HO-1 are further enhanced, the infarct size decreases and the brain edema improves [174]. Wu et al. [111] have demonstrated that, after stroke, curcumin administered by intraperitoneal injection (300 mg/kg) in rats inhibits oxidative stress, induces the expression of NQO1 and enhances the binding activity of Nrf2 to ARE. However, the PI3K/Akt pathway is necessary for curcumin effects, because blocking the PI3K/Akt signaling pathway abolishes the neuroprotective effects. To reduce the toxicity typically observed when curcumin is dissolved with DMSO or NaOH, Li et al. [169] administered curcumin dissolved in corn oil at 30 min after MCAO. The administration pre-reperfusion with curcumin reduced the subsequent IRI in the MCAO rat model as indicated by the reduction of brain edema, BBB disruption and neurological dysfunction at 24 h post reperfusion. The authors indicate that curcumin has significant neuroprotective effects after cerebral IRI by activating the Nrf2 pathway and by down-regulating NF-κB and MDA levels [169].

In hemorrhagic strokes, the lysis of red blood cells produces the release of hemin, a degradation product of hemoglobin. Hemin is a highly reactive compound and a dangerous molecule that is quickly accumulated and slowly degraded by HO, which causes damage in rat astrocytes and neurons [175]. The in vitro study by González-Reyes et al. [172] identifies curcumin as a neuroprotectant against hemin-induced damage in primary cultures of cerebellar granule neurons of rats. These in vitro data confirm that Nrf2 activation and antioxidant response (HO-1 and GSH) play a major role in the neuroprotective effect of curcumin. Although many experimental in vivo and in vitro studies have showed the protective effects of curcumin via the Nrf2 pathway, currently, no high-quality evidence showing that curcumin administration activates Nrf2 in humans is reported. The poor bioavailability of curcumin and its fast metabolism in humans are important factors to consider. Several approaches have been considered, including the adjuvant, the liposomal curcumin, curcumin nanoparticles and phospholipid complexes. The main structural modification of curcumin is to prepare the analogues without the β-diketone moiety, responsible for the instability and weak pharmacokinetic profiles of curcumin [176]. Furthermore, there is growing evidence that the addition of piperine may improve curcumin bioavailability [177,178]. Future well-controlled human intervention trials are needed to corroborate the neuroprotective effects of curcumin via the Nrf2 pathway observed in vitro and in animal studies of IRI and to advance our current understanding in humans.

#### 3.3.2. Fumarate

Dimethyl fumarate (DMF) is derived from the simple organic acid fumaric acid, which is named after the earth smoke plant (*Fumaria officinalis*). While free fumaric acid is poorly absorbed, DMF is rapidly metabolized to monomethyl fumarate (MMF) [179]. With a broad efficacy, good safety and satisfying tolerability, the compound is the first-line oral drug for multiple sclerosis disease [180] and its immunomodulatory potential is also explored in other immune-mediated diseases [181,182,183,184]. Pleiotropic biological effects characterize DMF, including anti-oxidative stress and anti-apoptotic and immunomodulatory properties, as well as providing protection from microvascular dysfunction in a variety of tissues [185]. In vitro experiments have shown that MMF prevents detrimental pro-inflammatory response promoting, in a dose-dependent manner, the polarization of T lymphocytes toward the T-helper cell type 2 (Th2) phenotype, a T cell subset characterized by the production of interleukin-(IL)4, IL-5 [186] and IL-10 [187]. This Th2 shift was later linked to direct effects of fumaric acid esters on dendritic cells (DCs), thus inducing functional type II DCs with in vivo relevant suppression of the proinflammatory cytokines IL-12 and IL-23 [188,189]. At high dosages, fumaric acid esters were also shown to induce apoptosis in vitro [190]. Beyond its effects on T-cells and dendritic cells, DMF may also target several other immunologically active cell types [187,191,192,193,194,195,196].

In addition to its modulatory effects on immune cells, DMF may also possess neuroprotective capacity. DMF inhibits the production of nitric oxide (NO), IL-1β, TNF and IL-6 in astrocytes and microglia, increases plasma levels of IL-10 and suppresses macrophage infiltration into the brain during autoimmune encephalomyelitis (EAE) [194]. Abundant evidence indicates that fumaric acid esters activate the Nrf2-Keap1 pathway and increase the natural antioxidant responses in vivo and in vitro [197,198]. DMF leads to direct modification of Keap1 [188,197] and can suppress NF-κB transcription, induces detoxification enzymes (i.e., GSH reductase, c-glutamylcysteine synthetase and GSH synthetase) in astrocytes and microglial cells and modulates glutathione levels in cells [199]. Recent data indicate that systemic DMF treatment is involved in maintaining BBB integrity and improving neurological outcomes in a short-term model of hemorrhagic strokes [121,200,201] and ischemic stroke [202,203]. In all of these cases, abundant evidence indicates that DMF/MMF act via activation of the Keap1-Nrf2-ARE signaling pathway [121,200,202]. Indeed, the beneficial effect of DMF was lost in the Nrf2-KO animals, suggesting that its therapeutic effect is mainly through activating Nrf2. However, the long-term neuroprotective effects observed after DMF treatment are also related to its immunomodulatory ability via an Nrf2-independent mechanism [125].

Liu et al. [204,205] have also provided evidence on the protection derived by the pretreatment with DMF against ischemic damage in initial, acute and extended phases after hypoxia-ischemia (HI). By using a cerebral HI mouse model and transgenic loss-of-function of Nrf2 mice, the authors have observed that pre-treatment with DMF for 7 days prior to hypoxia-ischemia confers robust and prolonged Nrf2-dependent neuroprotection by involving anti-oxidative and anti-inflammatory response and the attenuation of reactive gliosis in astrocyte and microglia. Overall, these findings support the unique protective role of Nrf2 in the stroke field and may open a new window to utilize these endogenous neuroprotection mechanisms as preventive approach in the development and progression of cerebral ischemia pathology.

#### 3.3.3. Resveratrol

Resveratrol (3,4,5-trihydroxystilbene) is a polyphenolic compound abundantly present in grapes and red wine [206]. It is well known for its antioxidant, anti-inflammatory and antiapoptotic properties [207,208,209,210,211] that it exerts by influencing multiple pathways [212]. Studies performed both in vitro and in vivo have provided evidence that resveratrol has neuroprotective effects. Resveratrol treatment of neuronal cell lines and hippocampal slice cultures exposed to oxygen and glucose deprivation (OGD)—a model of hypoxia/ischemia—promoted cell survival [213,214]. Recent studies demonstrated that resveratrol can protect hippocampal neurons from damage caused by transient cerebral ischemia [209]. In rodent models of ischemia, pre- and post-treatment with resveratrol determined a reduction of the infarct volume and brain edema [107,158], thus confirming the neuroprotective effects of this natural compound observed in in vitro models. Different mechanisms have been identified as responsible for the neuroprotective effects of resveratrol. Evidence exists demonstrating that it down-modulates the activity of the pro-apoptotic factors caspase-3 and Bax, promotes Bcl-2 expression and contrasts alteration of mitochondrial function [215,216], thus exerting anti-apoptotic effects. In a rat model of brain ischemia, it has been observed that resveratrol can induce neuroprotection by activating the PI3K/AKT signaling pathway, that has a key role in mediating cell survival, thus preventing neuronal death [211]. More recently, Hou et al. [217] deepened the understanding of the mechanisms involved in resveratrol-mediated neuroprotection using rats subjected to middle cerebral artery occlusion followed by reperfusion. They observed that pre-treatment with resveratrol for 7 days was able to reduce cerebral infarct area, neuronal damage and apoptosis and this was associated with increased expression of p-JAK2, p-STAT3, p-AKT and p-mTOR. The authors concluded that resveratrol is able to exert neuroprotective activity on cerebral IR by promoting the phosphorylation of key proteins of the JAK2/STAT3/PI3K/AKT/mTOR pathway. In vitro and in vivo studies also demonstrated an anti-inflammatory activity of resveratrol on activated microglia, as it effectively inhibits IL-1β, TNFα and nitric oxide production, together with NF-κB signaling and p38 phosphorylation [218,219,220,221], thus contrasting the deleterious effects of inflammation, an important factor involved in ischemic stroke. Another important property of resveratrol responsible for its neuroprotective effects is its antioxidant activity. It directly attenuates oxidative stress by scavenging ROS, thus inhibiting lipid peroxidation and DNA damage. In vitro and in vivo evidence exists demonstrating that the neuroprotective effects of resveratrol are due, at least in part, to its ability to activate the Keap1−Nrf2 pathway, which, in turn, modulates the expression of inflammatory mediators and of antioxidant enzymes [222,223]. Through the up-regulation of Nrf2 activity, resveratrol promotes the expression of ARE-regulated genes involved in the control of free radical levels [212]. In vitro experiments performed with neuronal cell lines and primary neuronal cells demonstrated that the activation of the Nrf2/ARE pathway by resveratrol promotes HO1 activity and the increase in glutathione and SOD levels [224,225]. The use of small interfering RNA in an in vitro oxidative stress model of endothelial cells showed that the antioxidant activity of resveratrol was inhibited if Nrf2 was knocked down [226]. In a recent study, Yang and colleagues [227] observed that in vitro resveratrol treatment of rat cortical neurons at different times reduced neuronal injury, decreasing lactate dehydrogenase and increasing SOD activity in a concentration-dependent manner. Cells treated with resveratrol showed increased cell viability and reduced apoptosis. The authors also observed that this treatment promoted the upregulation of Nrf2 and its translocation into the nucleus and the expression of NAD(P)H, NADPH quinone oxidoreductase 1 (NQO-1) and HO1, all of which are involved in contrasting oxidative stress. Of note, NQO-1 is able to reduce ROS levels, thus preventing cellular injury in brain ischemia and in neurodegenerative diseases such as Alzheimer’s disease, Parkinson’s disease and multiple sclerosis [228].

Studies conducted in rats demonstrated that pre-treatment with resveratrol up-regulated Nrf2 expression and increased HO1 levels after cerebral IRI [107]. Moreover, in a mouse model of cerebral ischemia, Narayanan et al. [151] showed that resveratrol-mediated neuroprotection was reduced in Nrf2^−/−^ mice, compared to wild type mice, thus demonstrating that resveratrol activity was Nrf2-dependent. These observations also confirm, in vivo, that the neuroprotective antioxidant activity of resveratrol is mediated, at least in part, by the activation of the Nrf2/ARE pathway. All these observations obtained using in vitro and in vivo models strongly sustain the therapeutic potential of resveratrol in ischemic cerebral damage. However, due to its rapid clearance from the circulation, further studies are needed to improve its efficacy in vivo.

#### 3.3.4. Sulforaphane

Sulforaphane (4-methylsulfinylbutyl isothiocyanate) is a natural isothiocyanate derived from the hydrolysis of glucoraphanin, widely present in cabbage, broccoli and other vegetables belonging to the family Brassicaceae [229,230]. It is a pleiotropic compound with anti-tumor and anti-microbial activities, as widely demonstrated in experimental models [231,232,233]. Data obtained in animal models showed a protective activity of sulforaphane in IRI affecting different tissues, including kidney [234], retina [235] and intestine [236,237].

Evidence exists demonstrating that it has also neuroprotective effects. Studies in rodents have shown that sulforaphane is able to reduce the cerebral infarct volume following focal ischemia and cerebral edema in injured brain [238,239]. Ma and colleagues [240] demonstrated that in vivo treatment with sulforaphane inhibited the NF-κB signaling pathway, thus reducing the expression of pro-inflammatory cytokines, nitric oxide and cyclooxygenase-2 in rats subjected to middle cerebral artery occlusion. Data obtained using in vitro and in vivo models evidenced that the protective effects of this compound are mainly due to its ability to activate the Nrf2/ARE pathway [241,242]. In a neonatal hypoxia-ischemia model, Ping and colleagues demonstrated that treatment with sulforaphane upregulated Nrf2 and HO1 expression and reduced neurons apoptosis and brain tissue loss [243]. Furthermore, in vitro administration of this compound in cultures of astrocytes, before or after exposure to oxygen–glucose deprivation conditions (OGD), improved cell survival by activating the Nrf2 pathway [244]. Sulforaphane promoted Nrf2 expression in cardiac cells and epidermal cells by the methylation of the Nrf2 promoter [245,246]. It also interacts with thiol groups of Keap1 cysteines, thus affecting the Nrf2/Keap1 complex stability and contrasting Nrf2 degradation [246,247,248]. Sulforaphane-mediated activation of the Nrf2 pathway induces antioxidative and detoxifying enzymes, such as glutathione S-transferase (GST), HO1 and NQO-1, that, in turn, play a crucial role in neuroprotection. Recently, in a rat model of vascular cognitive impairment, which involves the permanent occlusion of carotid arteries, it was demonstrated that administration of sulforaphane reduced ischemic injuries and improved cognitive abilities [249]. The observed neuroprotection was associated with increased Nrf2 activation and HO1 expression. To confirm the role of Nrf2 in sulforaphane-mediated protective effects, the same authors set up in vitro experiments with endothelial cells subjected to OGD conditions. They observed that, if Nrf2 was knocked down, sulforaphane was no longer able to protect endothelial cells from OGD-mediated damage, thus concluding that sulforaphane preserves the integrity of the blood–brain barrier via Nrf2 activation [249].

Evidence obtained both in humans and animal models has shown that this natural compound is rapidly absorbed and accumulated in tissues and that it is able to pass the blood–brain barrier and to accumulate in the brain [250,251,252,253,254]. All these observations suggest that sulforaphane could be a potential therapeutic molecule to treat cerebral ischemia injury.

#### 3.3.5. Tert-Butylhydroquinone

*Tert*-Butylhydroquinone (*t*BHQ) derives from the metabolism of the natural antioxidant compound butylated hydroxyanisole [88]. Several years ago, it obtained the approval for its use in humans [255,256] and it is widely used as a food additive. Studies conducted in different models of cerebral injury, including brain trauma and ischemic stroke, have demonstrated that it has neuroprotective effects [88,257]. In a rat model of IR, Shih et al. [88] observed that pretreatment with *t*BHQ reduced cerebral damage 24 h after stroke and it was associated with the increase of cortical GSH levels. The reduction of ischemic damage was observed even 1 month after and with both intracerebroventricular and intraperitoneal administration of *t*BHQ. Of note, *t*BHQ administration failed to induce cortical GSH increase and to reduce infarct size in Nrf2^−/−^ mice, thus suggesting that the neuroprotective activity of *t*BHQ is Nrf2-dependent [88]. These results have been confirmed in more recent papers. In a rat model of subarachnoid hemorrhage (SAH), Wang and collaborators [258] observed that *t*BHQ administration after SAH preserved blood–brain barrier integrity, as demonstrated by its ability to inhibit the increase of blood–brain barrier permeability evaluated by Evans blue extravasation. It also reduced cortical apoptosis and oxidative stress levels, neuronal degeneration and clinical behavior deficits. Moreover, significantly higher protein and mRNA expression levels of Nrf2, Keap1, HO1 and NQO1 were observed in animals treated with *t*BHQ, compared to those treated with vehicle [258], thus indicating the role of Nrf2 activation in *t*BHQ-mediated neuroprotection. Similar results were obtained in another recent study conducted using a rat model of neonatal hypoxic-ischemic encephalopathy [143]. The authors observed that post-treatment of animals with *t*BHQ reduced neuronal apoptosis in the cerebral cortex, infarct size and neuronal damage. The administration of this compound also improved neurological reflex, motor coordination and memory deficits. Furthermore, animals subjected to *t*BHQ administration showed higher levels of Nrf2 into the nucleus and increased expression of Nrf2-regulated antioxidative genes. All these data strongly support the neuroprotective effects of *t*BHQ and that these effects are mediated, at least in part, by the activation of the Nrf2 pathway. Due to its 1,4 diphenolic structure, *t*BHQ is able to dissociate the Nrf2/Keap1 complex, thus promoting the translocation of Nrf2 into the nucleus and the expression of antioxidant genes [259,260,261]. However, evidence exists showing that *t*BHQ also has detrimental effects [262], likely due to reactions mediated by GSH-conjugates [263]. Sun and colleagues [264] conducted a research study using a murine permanent middle cerebral artery occlusion model and observed that *t*BHQ treatment was associated with a significant increase in mortality, compared to control. They also observed that *t*BHQ significantly increased brain volume and impaired mitochondrial function of cerebrovascular endothelial cells, suggesting that *t*BHQ, by altering the blood–brain barrier integrity, can exacerbate stroke damage. Therefore, further studies are needed to determine whether *t*BHQ is able to promote long-term neuroprotection without severe side effects [143] to clarify its potential as therapeutic agent for stroke.

#### 3.3.6. Carbon Monoxide

The gaseous molecule carbon monoxide (CO) is associated with central nervous system toxicity. However, evidence also indicates that CO can be protective, depending on its concentration. CO is endogenously produced upon degradation of heme by HO. Heme oxygenase-1 (HO-1) participates in the cell defense against oxidative stress and is known to be induced by Nrf2 [265]. Zeynalov and Doré provided evidence in mice that CO can be therapeutic in IR brain injury [266] and its beneficial effect is mediated by activation of the Nrf2/Keap l/ARE/HO-1 pathway. Indeed, 250 ppm CO exposure promoted dissociation of Nrf2 from Keap1, increased the nuclear Nrf2 occupancy of AREs within the HO-1 promoter and induced time-dependent increases in HO-1 expression. Although the neuroprotection is completely lost in Nrf2^−/−^ mice, the beneficial effects of CO were also likely caused by the activation of other protective mechanisms. CO may also act through activation of anti-inflammatory, anti-apoptotic and vasodilatory mechanisms [267,268]. In addition, CO has been reported to have early thrombolytic effects after ischemia [269]. The ability to activate the Nrf2 signaling pathway and to pass through the blood–brain barrier (BBB), in concert with other anti-inflammatory mechanisms, render the low concentrations of inhaled CO an emerging good candidate for neurologic protection after stroke.

In addition to the aforementioned compounds, which represent the most studied Nrf2 activators with neuroprotective effects in experimental models of ischemic stroke, additional drugs (less frequently investigated compounds) have also shown beneficial effects, such as tanshinol borneol ester [270], Apelin 13 [141], ezetimibe [271], rosmarinic acid [100], biochanin A [272], isoquercetin [273], trilabatin [137], forsythiaside A [274], octreotide [275], Korean Red Ginseng [89], Schizandrin A [276], leonurine [277], sinomenine [101], myricetin [139], diterpene ginkgolides [278], totarol [279], paeonol [91], alpha-lipoic acid [280], omega-3 fatty acids [281], nicotinamide mononucleotide [282], chlorogenic acid [144], eriocitrin [283], bicyclol [114] andrographolide [284], phyllanthin [285], neohesperidin [286], protocatechualdehyde [287], osthole [275], salidroside [288], palmatine [289], pelargonidin [290] and britanin [291].

## 4. Challenges/Perspectives on the Use of Nrf2 Activators in Ischemic Stroke Patients

The scientific literature reviewed herein provides compelling evidence that Nrf2 activation is neuroprotective in ischemic stroke models. By using in vivo experimental approaches based mainly on the induction of permanent or transient cerebral ischemia in rodents, researchers have shown that both the endogenous and exogenously induced Nrf2 activations display neuroprotective effects. Concerning the endogenous activation, most of the studies presented in Table 1 (Section 3.2) clearly show the upregulation of Nrf2 and downstream targets following ischemia or IR. Of note, some studies show that Nrf2 knockout animals were more susceptible to cerebral ischemic stroke (higher infarct sizes and more severe neurological deficits), indicating that such endogenous activation has a major role in mitigating IR-related damage. Concerning the exogenously-induced activation of Nrf2, several well-known Nrf2 activators (described in Section 3.3) have displayed neuroprotective effects in ischemic stroke models and, notably, some Nrf2-activating compounds had superior neuroprotective effects against IRI in wild-type compared to Nrf2 knockout animals [89,151]. Thus, the reviewed literature provides an optimistic scenery and indicates that Nrf2 modulators may represent promising pharmacological strategies to treat ischemic stroke patients in a near future.

As already discussed in Section 2, the pro-oxidative events mediating IRI are diverse and may result from either increased production or decreased neutralization of oxidants, which are also diverse with respect to their chemical characteristics, including reactivity. Moreover, inflammation, which is closely related to oxidative stress, also represents a key event resulting from IR. In this scenery, it is reasonable to suppose that the treatment of acute stroke with Nrf2 activators could have advantages, compared to strategies based on a unique mechanism of action, such as (i) free radical scavengers aimed to neutralize a specific radical specie, or (ii) inhibitors of specific radical-generating enzymes (i.e., NOX). This is based on the fact that Nrf2 activation may lead to the upregulation of diverse players that counterbalance impairments in proteostasis, redox and inflammatory control [74]; their combined action might simultaneously mitigate distinct deleterious events resulting from IR.

On the other hand, there are significant drawbacks and challenges that work against the successful translation of the preclinical efficacy of Nrf2 activators into the clinical conditions of ischemic stroke patients. Initially, it is important to take into account the acute nature of ischemic stroke and the relative fast cascade of events resulting from the sudden loss of blood flow. As a consequence of the impaired delivery of oxygen and nutrients to cells, the brain’s electrical activity and signs of awareness have been reported to disappear within seconds after severe ischemic stroke, while energy stores seem to be depleted within minutes [292]. In such kind of situation, the hypothetical neuroprotective pharmacological treatment should be performed as soon as possible and, in addition, it is essential that it has a relatively quick pharmacological result in order to minimize neurodegeneration and related sequelae. In addition to the good pharmacokinetic properties, it is desired that the drug has a mechanism of action that allows the occurrence of neuroprotection even in metabolically compromised cells. As already discussed at the beginning of Section 3.1, Nrf2 is a transcription factor that controls the expression of cytoprotective genes [44,58,64,65,66,67,68,258,293,294] and is involved in different cellular processes [69,70]. mRNA and protein syntheses represent events that depend on the proper cellular metabolic homeostasis, which is compromised in ischemic cells [19]. In this context, although Nrf2 has been pointed as a therapeutic target for human chronic diseases [74], it seems that the translation of the beneficial effects of Nrf2 activators, observed in preclinical models, into clinical sceneries of ischemic stroke (an acute condition) is less probable. There are two important separate areas of the ischemic brain, the ischemic core and the ischemic penumbra. During a vessel occlusion, the core area is the first to be damaged, while cells from the ischemic penumbra are predominantly damaged during the reperfusion/reoxygenation period, thus representing a target for neuroprotection shortly after ischemic stroke episodes [295]. Taking into account that the beneficial effects of Nrf2 activators commonly depend on transcription and translation, it is likely that their potential neuroprotective effects in clinical stroke (if any) are restricted to cells located far from the ischemic center, whose metabolic compromise is not sufficient to impair transcription and translation. In this regard, it is important to recapitulate the experimental study by Takagi and collaborators, which showed increased levels of Nrf2 in the cerebral cortex and striatum of mice subjected to transient middle cerebral artery occlusion; notably, such increase was observed particularly in the penumbra zone [82], which is less metabolically compromised compared to the ischemic core. This experimental evidence reinforces the idea that metabolically impaired cells located near the ischemic core are unable to properly upregulate Nrf2 and related downstream proteins. This represents a conceivable disadvantage in using Nrf2 activators to treat extreme acute metabolic impairments, such as severe ischemic stroke.

This supposed disadvantage of using Nrf2 activators to treat acute conditions is in line with the idea that *timing* is crucial in modulating Nrf2 in disease [64]. Within this panorama, the literature reviewed herein indicates a significant number of studies using pretreatments when evaluating neuroprotective effects of Nrf2 activators in models of experimental ischemic stroke. Considering the acute nature of ischemic stroke, translating experimental results on Nrf2 activators into real clinical conditions seems to be less likely when data are derived from protocols based on pretreatments. This seems to be particular important considering that (i) Nrf2-derived biological effects are greatly dependent on transcription and translation and (ii) acute ischemic stroke leads to quick cell metabolic impairment. The design of protocols that properly mimic the real conditions of ischemic stroke patients (i.e., post-treatment with drugs after the diagnosis of stroke) will certainly maximize the possibility of progression of Nrf2 activators from bench to clinical conditions if the aim is to treat (not prevent) ischemic stroke.

In summary, the literature reviewed herein has unequivocally shown neuroprotective effects of the exogenously induced Nrf2 activation in experimental models of ischemic stroke, providing a positive panorama and indicating that Nrf2 modulators may represent promising pharmacological strategies to treat ischemic stroke patients in a near future. However, the acute nature of ischemic stroke represents a challenge when using a putative pharmacological strategy (Nrf2 activator) that relies upon transcription, translation and metabolically active cells. In this context, the execution of experimental protocols able to mimic real conditions of ischemic stroke patients in order to progress Nrf2 activators from preclinical studies to clinical practices seems crucial.

## Figures and Tables

**Figure 1 molecules-26-05001-f001:**
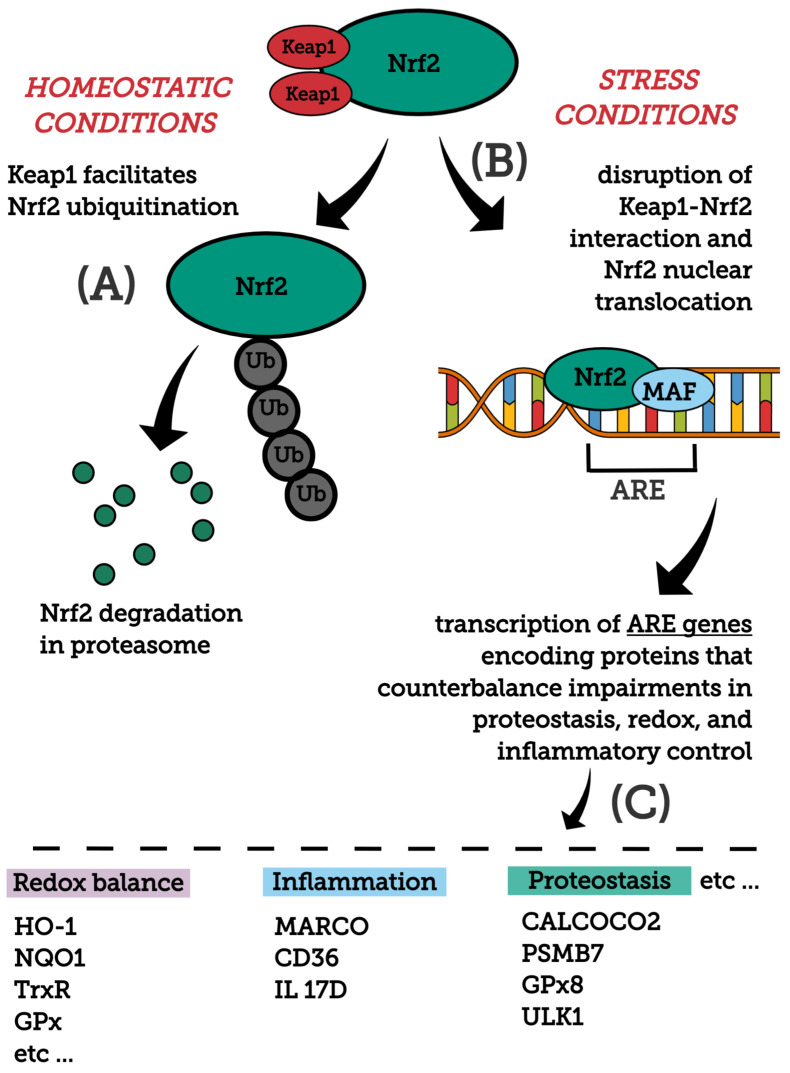
General overview of the Nrf2 pathway. Major molecules and events modulating Nrf2 stability and activation, as well as main downstream protein targets and their functions are shown. (**A**) Nrf2 structure is composed by domains (i.e., Neh2 and Neh6) that, after redox or signaling regulation, affect Nrf2 stability. Under homeostatic conditions, Keap1 binds to Nrf2, directing this transcription factor to ubiquitination and subsequent degradation by the proteasome. Keap1 is a redox sensor that, upon oxidative thiol modification, loses its capability to repress Nrf2. Glycogen synthase kinase 3 (GSK-3)-mediated phosphorylation of Nrf2 represents an alternative mechanism, facilitating its ubiquitination and consequent degradation by the proteasome. (**B**) Under stress conditions (excessive accumulation of ROS, electrophilic molecules and proteotoxic stress), Nrf2-Keap1 interaction is disrupted and Nrf2 translocates into the nucleus, wherein it heterodimerizes with small Maf proteins (sMaf) and binds to an enhancer sequence termed ARE that is present in the regulatory regions of over 250 genes (ARE genes). (**C**) These ARE genes, whose encoded proteins participate in diverse cellular/metabolic events, display significant roles in counteracting imbalances in proteostasis, redox and inflammatory control. For a detailed review on the mechanisms mediating Nrf2 stability and activation, see [74]. CALCOCO2, calcium binding and coiled-coil domain 2; CD36, CD36 scavenger receptor; GPx, glutathione peroxidase; Gpx8, glutathione peroxidase 8; HO-1, heme oxygenase-1; IL 17D, interleukin-17D; NQO1, NADPH Quinone oxidoreductase enzyme; PSMB7, proteasome subunit b type-7; TrxR, thioredoxin reductase; ULK1, unc-51 like autophagy activating kinase 1.

**Figure 2 molecules-26-05001-f002:**
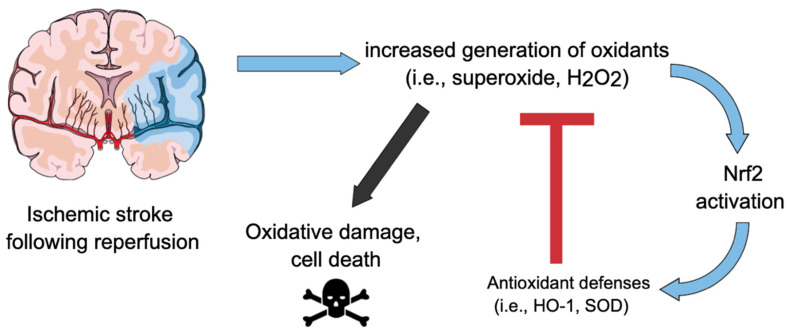
Interplay between ischemic stroke and Nrf2. Ischemia-reperfusion leads to increased levels of oxidants (i.e., hydrogen peroxide (H_2_O_2_), superoxide anion (O_2_^•−^)) that, in turn, can activate Nrf2. Nrf2-downstream proteins (i.e., heme oxygenase-1 (HO-1), superoxide dismutase (SOD)) can mitigate the deleterious effects of oxidants produced in excess during IR, preventing oxidative stress and cell death.

**Table 1 molecules-26-05001-t001:** In vivo experimental studies reporting the endogenous modulation of Nrf2 and/or Nrf2-downstream proteins after cerebral ischemia or ischemia-reperfusion.

Species (Sex; Age)		Experimental Model	#Findings Indicating Endogenous Modulation of Nrf2 after Ischemic Stroke		Tissue	Ref.
			Specific Findings	General Effect		
MICE	ICR mice (M; 8 weeks)	MCAO/R (1 h/2, 8, 24, 72 h)	↑ Nrf2, HO-1 and Trx protein expression ↓ Keap1 protein expression	↑	ischemic brain tissue	[90]
	ddY mice (M; 8–12 weeks)	MCAO/R (1 h/6, 24, 48 h)	↑ Nrf2 and HO-1 protein expression	↑	ischemic cortex and striatum	[82]
	CD-1 mice (M; NI)	pMCAO (24 h)	↑ nuclear Nrf2 translocation ↑ HO-1 protein expression ↑ SOD activity	↑	ischemic cerebral cortex	[91]
	ICR mice (M; 8–10 weeks)	MCAO/R (1 h/24 h)	↑ Nrf2 and HO-1 mRNA levels ↑ nuclear Nrf2 and HO-1 protein expression	↑	ischemic cerebral cortex	[92]
	OKD48 mice (M, F; 11–13 weeks)	MCAO/R (45 min/ 12, 24, 72 h, 7 d)	↑ Nrf2 and HO-1 protein expression	↑	peri-ischemic brain tissue	[93]
	Mice (NI) (M; 12 weeks)	MCAO/R (2 h/24 h)	↑ Nrf2, HO-1, GCL protein expression	↑	ischemic brain tissue	[94]
	C57BL/6 mice (M; 8 weeks)	BCCAO/R (20 min/24 h)	↑ Nrf2 DNA binding activity ↑ nuclear Nrf2, HO-1 and NQO1 protein expression	↑	striatum	[95]
	C57BL/6J mice (M; 8–10 weeks)	MCAO/R (1 h/24 h)	↑ Nrf2 and HO-1 protein expression	↑	ischemic brain tissue	[96]
	ICR mice (M; 6 weeks)	MCAO/R (1 h/24–72 h, 7 d)	↑ Nrf2 protein expression ↓ Keap1 protein expression	↑	ischemic brain tissue	[97]
	C57BL/6 mice (M; 10–18 weeks)	pMCAO (1 d–3 d)	↑ NQO1, HO-1, SOD2 and GPX-1 protein expression	↑	ischemic brain tissue	[89]
	C57BL/6 mice (M; 10–12 weeks)	MCAO/R (1.5 h/7 d)	↑ Nrf2 and SOD1 protein expression	↑	ischemic brain tissue	[98]
	ddY mice (M; 5–8 weeks)	MCAO/R (2 h/2, 6, 22 h)	↑ Nrf2 and HO-1 protein expression	↑	ischemic brain tissue	[99]
	CD-1 mice (M; 4 weeks)	MCAO/R (1 h/24 h)	↑ Nrf2 and HO-1 mRNA ↑ Nrf2 and HO-1 protein ↑ HO-1 activity	↑	ischemic brain tissue	[100]
	C57BL/6 mice (M; 12 weeks)	MCAO/R (1 h/3 d)	↑ nuclear Nrf2, HO-1 and NQO1 protein expression	↑	cerebral tissue	[101]
	ICR mice (M; NI)	MCAO/R (2 h/24 h)	↑ cytosolic Nrf2, HO-1 and NQO1 protein expression	↑	hippocampus	[102]
	BALB/c mice (M; 7 weeks)	MCAO/R (1.5 h/24–72 h)	↑ Nrf2, HO-1 and iNOS protein expression ↓ SOD activity	↑	ischemic brain tissue	[103]
	ICR mice (M; 8 weeks)	p dMCAO (7 d)	↑ Nrf2 protein expression	↑	ischemic cerebral cortex	[104]
	C57BL/6J mice (M; NI)	MCAO/R (1.5 h/24 h)	↓ SOD 2, HO-1, NQO1 and Nrf2 protein expression ↓ SOD activity ↑ NADPH oxidase protein expression	↓	ischemic brain tissue	[105]
	C57BL/6 mice (M; NI)	MCAO/R (1 h/72 h)	↓ Nrf2 protein expression	↓	hippocampus and cerebral cortex	[106]
RATS	SD rats (M; adult)	MCAO/R (2 h/2, 6, 24 h)	↑ Nrf2 and HO-1 protein expression ↑ Nrf2 and HO-1 mRNA levels	↑	ischemic cerebral cortex	[107]
	SD rats (M; NI)	pMCAO (72 h)	↑ Nrf2 and HO-1 protein expression	↑	ischemic cerebral cortex	[108]
	SD rats (M; adult)	pMCAO (72 h)	↑ Nrf2 and HO-1 protein expression	↑	ischemic brain tissue	[109]
	SD rats (M; NI)	MCAO/R (70min/ 4, 24, 72 h)	↑ Nrf2 and HO-1 protein expression	↑	ischemic brain tissue	[110]
	SD rats (M; 9 weeks)	MCAO/R (1 h/24 h)	↑ Nrf2 and NQO1 protein expression ↑ Nrf2 binding activity to ARE	↑	ischemic brain tissue	[111]
	SD rats (M; adult)	MCAO/R (2 h/22 h)	↑ Nrf2 (nuclear) and HO-1 protein expression	↑	cerebral cortex	[112]
	SD rats (M; adult)	MCAO/R (2 h/24 h)	↑ Nrf2 protein expression ↑ HO-1 protein expression	↑	ischemic brain tissue	[113]
	SD rats (M; NI)	pMCAO (24 h)	↑ nuclear Nrf2 translocation ↑ HO-1 protein expression ↑ Nrf2 and HO-1 mRNA levels	↑	ischemic brain tissue	[114]
	SD rats (M; NI)	pMCAO (24 h)	↑ Nrf2 and HO-1 protein expression	↑	ischemic brain tissue	[115]
	SD rats (M; adult)	MCAO/R (1 h/ 72 h)	↑ Nrf2 and NQO1 protein expression	↑	ischemic cerebral cortex	[116]
	Wistar rats (M; NI)	MCAO/R (1 h/72 h)	↑ Nrf2, HO-1 and NQO1 mRNA levels	↑	ischemic brain tissue	[117]
	Wistar rats (M; 6 months)	BCCAO/R (30 min/72 h)	↑ Nrf2 and HO-1 protein expression	↑	hippocampus	[118]
	SD rats (M; NI)	pMCAO (72 h)	↑ Nrf2 and HO-1 protein expression	↑	ischemic brain tissue	[119]
	SD rats (M; 10–12 weeks)	MCAO/R (1 h/24 h)	↑ Nrf2, Txr-1, Prdx1, Prdx2, Prdx3 and Prdx4 protein expression ↑ Nrf2, Txr-1, Prdx1, Prdx2, Prdx3 and Prdx4 mRNA levels	↑	ischemic brain tissue	[120]
	SD rats (M; NI)	MCAO/R (2 h/24 h)	↑ Nrf2 protein expression ↑ Nrf2 mRNA levels ↓ SOD activity	↑	ischemic brain tissue	[121]
	SD rats (NI; NI)	BCCAO/R (10 min/1–7 d)	↑ Nrf2 and HO-1 protein expression	↑	ischemic brain tissue	[122]
	SD rats (M; adult)	MCAO/R (1.5 h/24 h)	↑ Nrf2 and HO-1 protein expression ↑ Nrf2 and HO-1 mRNA levels	↑	hippocampus	[123]
	SD rats (M; adult)	pMCAO (24 h)	↑ nuclear Nrf2 translocation ↑ HO-1 and SOD1 protein expression ↑ HO-1 mRNA levels ↑ SOD1 activity	↑	ischemic cerebral cortex	[124]
	SD rats (M; adult)	MCAO/R (2 h/ 72 h)	↑ Nrf2 protein expression ↑ HO-1 protein expression	↑	ischemic brain tissue	[125]
	SD rats (M; NI)	MCAO/R (2 h/7 d)	↑ Nrf2 protein expression	↑	ischemic brain tissue	[126]
	SD rats (M; adult)	MCAO/R (1 h/24 h)	↑ Nrf2, NQO1 and Srnx1 protein expression ↑ Prdx 1, Prdx 2, Prdx 3, Prdx 4 protein expression	↑	ischemic brain tissue	[127]
	Hannover-Wistar rats (M; NI)	MCAO/R (1 h/24 h)	↑ Nrf2 protein expression ↑ SOD and GPx activity	↑	hippocampus	[128]
	Wistar rats (M; adult)	BCCAO/R (45 min/24 h)	↑ iNOS and Nrf2 protein expression	↑	hippocampus	[129]
	SD rats (M; 60–80 days)	MCAO/R (1 h/24 h)	↑ Nrf2 and Trx1 mRNA levels ↑ Trx1 protein expression	↑	ischemic brain tissue	[130]
	SD rats (F; adult)	MCAO/R (1.5 h/72 h)	↑ Nrf2 and NQO1 protein expression	↑	ischemic brain tissue	[131]
	Wistar rats (M; adult)	MCAO/R (1.5 h/72 h)	↑ Nrf2, HO-1 and NQO1 mRNA levels	↑	ischemic brain tissue	[132]
	SD rats (M; adult)	PCI (20 min)	↑ Nrf2 and HO-1 protein expression	↑	cerebral cortex ischemic penumbra	[133]
	SD rats (M; NI)	Focal PTI	↑ Nrf2 and HO-1 protein expression	↑	penumbra of cerebral infarction	[134]
	SD rats (M; 7–10 weeks)	MCAO/R (2 h/72 h)	↑ Nrf2 and HO-1 protein expression	↑	cerebral cortex and striatum	[135]
	SD rats (M; NI)	MCAO/R (1 h/6 or 24 h)	↑ Nrf2 protein expression ↓ SOD activity	↑	ischemic penumbra	[136]
	SD rats (M; Adult)	MCAO/R (2 h/72 h)	↑ nuclear Nrf2 protein expression ↓ cytosolic Nrf2, NQO1 and HO-1 protein expression ↓ SOD activity	↑	ischemic brain tissue	[137]
	SD rats (M; 8 weeks)	MCAO/R (2 h/24 h)	↑ HO-1 protein expression ↓ Nrf2 and Trx protein expression	↑↓	cerebral cortex	[113]
	SD rats (M; adult)	MCAO/R (2 h/72 h)	↓ SOD activity ↓ Nrf2, HO-1 and NQO1 protein expression	↓	ipsilateral ischemic tissue	[138]
	SD rats (M; 10 months)	MCAO/R (2 h/48 h)	↓ HO-1 protein expression ↓ SOD activity	↓	ipsilateral ischemic tissue	[139]
	Wistar rats (M; NI)	MCAO/R (2 h/24 h)	↓ GPx and SOD activity ↓ Nrf2 and NQO1 protein expression	↓	ischemic brain tissue	[140]
	SD rats (M; 3 months)	MCAO/R (1.5 h/72 h)	↓ NQO1, HO-1 and cytoplasmic Nrf2 protein expression	↓	ischemic brain tissue	[141]
	SD rats (M; 7–8 weeks)	MCAO/R (1.5 h/7 d)	↓ Nrf2 mRNA levels	↓	ischemic brain tissue	[142]
	SD rats (M; NI)	MCAO/R (2 h/7 d)	↓ SOD activity ↓ nuclear Nrf2, HO-1 and NQO1 protein expression ↓ Nrf2, HO-1 and NQO1 mRNA levels	↓	cerebral cortex	[143]
	SD rats (NI; NI)	BCCAO/R (10 min occlusion + 10 min reperfusion + 10 min occlusion)	↓ Nrf2, NQO1 and HO-1 protein expression ↓ SOD activity	↓	ischemic brain tissue	[144]
	SD rats (F; NI)	MCAO/R (1 h/24 h)	↓ Nrf2 protein expression	↓	hippocampus	[145]
	SD rats (M; NI)	MCAO/R (1.5 h/14 d)	↓ Nrf2 protein expression ↑ Keap-1 protein expression ↓ SOD and catalase activities	↓	ischemic brain tissue	[146]

*Abbreviations and symbols:* ↑, activation. ↓, inhibition. M, male; F, female. SD rats, Sprague-Dawley rats. NI, not informed. BCCAO, bilateral common carotid arteries occlusion. MCAO, middle cerebral artery occlusion. MCAO/R, middle cerebral artery occlusion followed by reperfusion. PCI, photochemical cerebral ischemia. pdMCAO, permanent distal middle cerebral artery occlusion. pMCAO, permanent middle cerebral artery occlusion. PTI, photothrombotic ischemia model. ARE, antioxidant response element. HO-1, heme oxygenase 1. NQO1, NAD(P)H:quinone oxidoreductase 1. Prdx, peroxiredoxin. SOD, superoxide dismutase. Srnx1, sulfiredoxin-1. Trx1, thioredoxin. *Notes:* # Only significant differences between control/sham and ischemic animals were evaluated. HO-1, Heme oxygenase 1: Nrf2-downstream protein catalyzing the degradation of heme, producing biliverdin, ferrous iron and carbon monoxide. NQO1, NAD(P)H:quinone oxidoreductase 1: Nrf2-downstream protein catalyzing the two-electron reduction of quinones and a wide range of other organic compounds. Its physiological role is partly related to the reduction of free radical load in cells and the detoxification of xenobiotics. Prdxs, peroxiredoxins: ubiquitous family of Nrf2-downstream antioxidant enzymes involved in the reduction of peroxides (specifically hydrogen peroxide). SODs, superoxide dismutases: enzymes catalyzing the dismutation of the superoxide radical into molecular oxygen and hydrogen peroxide. Srnx, Sulfiredoxin: Nrf2-downstream protein member of the oxidoreductases family catalyzing reduction of oxidative modifications (i.e., sulfinic, disulfides, etc.). Trx, Thioredoxin: class of small redox (Nrf2-downstream) proteins playing important roles in redox signaling.

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
