# Peer review of "The Nrf2 Pathway in Ischemic Stroke: A Review"

_molecules, 2021, doi:10.3390/molecules26165001_

Round 1

Reviewer 1 Report

The authors wrote a complete review about the the protective role of nuclear factor erythroid 2-related factor 2 (Nrf2) in mitigating the oxidative events for ischemic stroke treatment. I believe this review will attract researchers’ interest in stroke treatment, and this review fits to the journal expectations. In my opinion, it can be accepted after several minor changes:

  1. There are many neuroprotectants for ischemic stroke treatment to protect the brain from oxidative damage (e.g. free radical scavengers, NADPH oxidase (NOX) inhibitors, et al.). The authors should consider about comparing Nrf2 activator with those neuroprotectants in the introduction or discussion section to highlight the advantage of Nrf2 activator.
  2. Please make sure the brain scheme in Figure 1 is original. There seems to be similar images somewhere else (https://www.eurekaselect.com/176534/article).

Reviewer 2 Report

The manuscript from Farina and co-workers is a well-written review on the participation of the transcription factor Nrf2 in brain ischemic stroke. This protein regulates the expression of several antioxidant proteins and it is clearly important to handle different oxidative stress scenarios. The review aimed to summarize the current experimental evidence which suggested Nrf2 as protective in ischemic stroke as well as highlighting it as a putative drug target to alleviate deleterious oxidative events. Authors also described some of the molecular mechanisms behind the Nrf2 pathway functioning and discussed the potential use of different Nrf2 activators in ischemic stroke therapy.

In my opinion the review is quite complete and describes in a clear way all the intended topics. Although I have no major issues with this manuscript, I still have a couple of comments for the authors; therefore, making this review not ready to be accepted in its present form. I hope my suggestions would be helpful to improve the final version though.

Specific comments:

  • Table 1 is hard to read because it includes substantial information in the same columns and is not easy to visualize immediately. Besides, the third column shows too much text that can be shown in the legend and not directly matching to what we see in the rows; i.e., tissue/injury. My suggestion would be to separate the information that is shown between parentheses in independent columns and if the chronological order is not necessary for the purposes of this review, showing all findings from a given species consecutively for an easier reading and comparison. Similar or redundant findings could be combined in the same row and showing multiple references in the last column. If allowed, you could also show the upregulation/downregulation as color-coded text for simplicity (optional).

  • An additional figure showing the full Nrf2 pathway, protein targets and the cell compartments involved in this regulatory network should be included to better complement and illustrate sections 2 and 3.

  • Regarding acronyms and abbreviations, please double-check the consistency of these when mentioned in the text. I have seen different acronyms referring to the same thing. For example, IR injury or I/R injury instead of IRI (ischemic reperfusion injury) in lines 379, 394, 414, 517. Please use only one acronym/abbreviation throughout the manuscript.

  • In line 511, NQO-1 should be NAD(P)H quinone oxidoreductase 1… it reads “quinine”.
